# The Role of microRNA in Spermatogenesis: Is There a Place for Fertility Preservation Innovation?

**DOI:** 10.3390/ijms25010460

**Published:** 2023-12-29

**Authors:** Charlotte Klees, Chrysanthi Alexandri, Isabelle Demeestere, Pascale Lybaert

**Affiliations:** 1Research Laboratory on Human Reproduction, Faculty of Medicine, Université Libre de Bruxelles (ULB), 1070 Brussels, Belgium; charlotte.klees@ulb.be (C.K.); chrysanthi.alexandri@ulb.be (C.A.); isabelle.demeestere@ulb.be (I.D.); 2Fertility Clinic, HUB-Erasme Hospital, 1070 Brussels, Belgium

**Keywords:** MicroRNA, fertility preservation, spermatogenesis, gonadotoxicity, infertility

## Abstract

Oncological treatments have dramatically improved over the last decade, and as a result, survival rates for cancer patients have also improved. Quality of life, including concerns about fertility, has become a major focus for both oncologists and patients. While oncologic treatments are often highly effective at suppressing neoplastic growth, they are frequently associated with severe gonadotoxicity, leading to infertility. For male patients, the therapeutic option to preserve fertility is semen cryopreservation. In prepubertal patients, immature testicular tissue can be sampled and stored to allow post-cure transplantation of the tissue, immature germ cells, or in vitro spermatogenesis. However, experimental techniques have not yet been proven effective for restoring sperm production for these patients. MicroRNAs (miRNAs) have emerged as promising molecular markers and therapeutic tools in various diseases. These small regulatory RNAs possess the unique characteristic of having multiple gene targets. MiRNA-based therapeutics can, therefore, be used to modulate the expression of different genes involved in signaling pathways dysregulated by changes in the physiological environment (disease, temperature, ex vivo culture, pharmacological agents). This review discusses the possible role of miRNA as an innovative treatment option in male fertility preservation–restoration strategies and describes the diverse applications where these new therapeutic tools could serve as fertility protection agents.

## 1. Introduction

The preservation of male fertility is crucial in cancer patients who undergo cytotoxic treatments, such as chemotherapy or radiation therapy, that can impair spermatogenesis and lead to infertility [1]. The most established and non-invasive option for fertility preservation in adults and post-pubertal boys is semen cryopreservation, which allows for the long-term storage of sperm for future use. Additionally, innovative approaches like immature testicular tissue (ITT) cryopreservation and spermatogonial stem cell (SSC) transplantation have demonstrated promising results in preserving fertility for prepubertal boys, but they are still considered to be experimental techniques with safety limitations [2]. Given that sperm cryopreservation is not possible before puberty, and that the prepubertal testicular environment is more sensitive to oncological treatment compared to that of adults, there is an urgent need to develop novel strategies to safeguard the fertility of childhood cancer patients [3].

Over the last decade, microRNAs (miRNAs) have emerged as promising molecular markers and therapeutic tools in a variety of diseases. Their role as fertility protective agents in male fertility preservation–restoration strategies is a new and innovative therapeutic option that is currently being investigated [4]. Indeed, the integration of miRNAs into existing male fertility preservation strategies offers a promising avenue for enhancing the effectiveness and improving the outcomes of these interventions. MiRNAs can be used to target specific molecular pathways and enhance the survival and quality of preserved sperm or testicular tissue. Hence, miRNAs can be proposed as an option for optimizing the cryopreservation process or to potentially address limitations in current fertility preservation techniques, such as low post-thaw viability or reduced functional capacity of preserved samples. Moreover, the use of miRNAs may hold promise in improving SSC transplantation for optimizing cell survival, promoting functional integration, and improving the overall success of the procedure.

The unique characteristics of miRNAs render them potential candidates in the development of less-invasive fertility preservation approaches through pharmacological protection of spermatogenesis [5]. It is well known that a single miRNA can have multiple gene targets and, therefore, miRNA-based therapeutics can be used to modulate the expression of different genes involved in signaling pathways that become dysregulated during oncological treatment such as apoptosis, DNA damage responses, and cell proliferation. Additionally, miRNAs present not only tissue-specific but also cell-specific expression patterns, providing an opportunity for selective targeting. MiRNAs play important roles in various aspects of spermatogenesis, including germ cell development, meiosis, and sperm maturation [6]. Alterations in miRNA expression profiles have been observed in response to gonadotoxic therapies, while manipulation of miRNA expression through exogenous delivery or inhibition approaches has shown promising results in experimental models [7,8]. At the same time, the development of efficient and targeted delivery systems, such as nanoparticle-based carriers or exosomes, will enable the precise delivery of miRNAs to the desired cells or tissues within the male reproductive system. Therefore, understanding the regulatory functions of miRNAs in male fertility and exploring their clinical applications holds significant promise for improving the success of fertility preservation strategies in male cancer survivors.

The extent of testicular damage associated with cancer treatment varies depending on the drug type and dosage. For individuals with Hodgkin’s disease (HD), for example, the likelihood of developing azoospermia differs based on the treatment regimen. Specifically, it has been reported that 97% of men treated with MOPP (mechlorethamine, vincristine, procarbazine, and prednisone) experience azoospermia, while the rate of azoospermia is 54% for those treated with ABVD (adriamycin, bleomycin, vinblastine, and dacarbazine) [9]. In a survey conducted by Benedict et al. in 2016, 43 individuals who had experienced cancer during their adolescent and young adult years shared their perspectives on fertility. The findings revealed that 50% of male respondents and 39% of female respondents expressed concerns regarding uncertainty about their fertility [10]. Consequently, the potential adverse effects of cancer treatments that result in diminished fertility pose a significant challenge to the future well-being of cancer patients.

The goal of this review is to discuss existing fertility preservation methods in the context of the possible implementation of miRNAs as new therapeutic tools based on the ability of these versatile molecules to modulate gene expression across dysregulated signaling pathways during gonadotoxic treatment. The review explores the diverse applications, benefits, and challenges to the use of miRNAs in this setting and also highlights the role of miRNAs as innovative molecular markers.

## 2. The Characteristics of miRNAs

The discovery of miRNAs dates back over three decades to Lee et al. when *lin-4* was described as a small non-coding RNA in the nematode *Caenorhabditis elegans*. Interestingly, the gene *lin-4* did not code for a protein but instead produced a pair of small RNAs responsible for the downregulation of the LIN-14 protein, which serves as a repressor, controlling the transition from the L1 larval stage to the L2 stage [11,12]. A second small RNA was described soon after, let-7a, which is also involved in developmental regulation in *C. elegans* and is highly conserved across animal species [13,14]. According to miRbase, over 1900 and 1200 miRNAs have, since their first description, been identified in humans and mice, respectively [6].

MiRNAs are endogenous small non-coding RNA molecules with a length of 21 to 25 nucleotides that regulate gene expression at the post-transcriptional level by binding to a target messenger RNA (mRNA), most commonly through its 3′ untranslated region (3′UTR) [15,16]. The fate of the target mRNA depends, among other parameters, on the complementarity between the miRNA and its target, potentially resulting in either degradation or translational inhibition, impeding protein synthesis [17]. Each miRNA can have many targets, and each mRNA can be targeted by several miRNAs [6]. These features make miRNAs major post-transcriptional gene regulators, and an estimated one-third of the human genome is regulated by miRNAs [18]. Furthermore, miRNAs are known to be well-conserved across different species [19]. The use of bioinformatics has become a valuable tool for identifying miRNAs, as well as for predicting their target genes and exploring their functions across different species due to their well-conserved nature [20]. Several databases, including mirBase, miRtarbase, TargetScan, and TarBase, are now available to provide information about experimentally validated and predicted miRNA targets and, therefore, to elucidate the involved gene pathways [5].

MiRNAs are known to regulate important biological processes [15]. Aberrant expression of miRNAs has also been shown to be associated with many diseases, and as such, these molecules can be used as diagnostic and prognostic biomarkers [21,22].

Depending on the location of the miRNA coding region in the genome, which can be either inter- or intra-genic, miRNAs are processed by a canonical or non-canonical pathway. The canonical pathway is the one most commonly used during miRNA biogenesis [23] (Figure 1).

Starting in the nucleus, the miRNA coding region is first transcribed by an RNA-polymerase II into a long stem-loop pri-miRNA and cleaved into a precursor-miRNA (pre-miRNA) by a microprocessor complex [24]. This complex is formed by the combination of a ribonuclease III enzyme called DROSHA along with the co-factor DGCR8, which recognizes the RNA substrate [16]. The pre-miRNA is taken by a protein called Exportin-5 from the nucleus to the cytoplasm in a Ran-GTPase-dependent manner [24,25]. The pre-miRNA is then cleaved into a mature miRNA duplex by another RNase III enzyme called DICER [25]. This duplex is incorporated into the RNA-induced silencing complex (RISC) together with an Argonaute protein (AGO). Only one strand is selected, the guided strand, and the passenger strand is commonly degraded [26,27]. The seed sequence (nucleotides 2–7 in the 5′ region) of the mature miRNA can now bind to the targeted mRNA, generally in its 3′ UTR [28]. This binding leads to the degradation of the mRNA or its translational inhibition, depending on their complementarity.

Alternative pathways, the non-canonical miRNA biogenesis pathways, such as DROSHA/DCGR8-independent and DICER-independent pathways, have also been described [16].

## 3. The Role of miRNAs in Spermatogenesis

To discuss the possible application of miRNA-based therapies to fertility preservation strategies, it is essential to understand the physiological role of miRNAs throughout male gamete production and pathophysiological dysregulation in diseases of the reproductive organs. Spermatogenesis is a complex process that takes place within the seminiferous tubules of the testes to produce mature haploid spermatozoa from diploid SSCs. Within the seminiferous tubules, a population of germ cells undergoes a complex differentiation process, which is facilitated and supported by Sertoli cells. Outside the tubules, the Leydig cells are responsible for hormone synthesis. Spermatogenesis consists of three different phases: the mitotic division, the meiotic division, and spermiogenesis. During the first phase, spermatogonia undergo mitotic divisions to assure self-renewal and maintain the pool of SSCs or to enter a differentiation pathway. During the meiotic phase, spermatogonia start by duplicating their DNA to generate primary spermatocytes [29]. This is then followed by two meiotic divisions. The first division takes place to generate secondary spermatocytes, and the second produces haploid spermatids. The haploid spermatids then undergo a transformation called spermiogenesis, involving numerous morphological changes (acrosomal and flagellum development, chromatin condensation, and cytoplasmic condensation and exclusion), to produce mature spermatozoa, which are released into the lumen of seminiferous tubules [30,31]. This highly complex mechanism depends not only on hormonal factors and metabolic regulation but also on the strict regulation of genes by transcriptional, post-transcriptional, and epigenetic processes [31]. All stages of the spermatogenesis process are regulated by numerous transiently expressed protein factors. This explains why each differentiating cell type has its own transcriptome profile [30]. Several studies have highlighted the role of small RNAs (siRNAs, piRNAs, miRNAs) in spermatogenesis and, therefore, in male fertility (reviewed in [6,17,29,32,33,34,35,36,37,38]). In particular, studies have shown that miRNAs can be expressed exclusively or preferentially in the testis and in specific testicular cell types, as shown by Gan et al., who found miRNAs to be more abundant in spermatogonia than in other testis cell types [34,39,40,41]. In addition to specific expression in male reproductive tissues, the role of miRNA has also been highlighted in steroidogenesis and in zygotic and early embryo reprogramming [35].

The contributions of miRNA to spermatogenesis and testicular function have been demonstrated as they have been implicated in the regulation of proliferation and differentiation of both Sertoli and germ cells [6] (Figure 2). The induced inactivation of enzymes involved in the biosynthesis of miRNAs leads to their disruption and demonstrates their essential role in germ cell survival and proper development of spermatogenesis, causing fertility disorders in males [33,35]. A deletion in DICER in mouse primordial germ cells or spermatogonia causes defects in spermatogonial progression and differentiation, lower numbers of haploid cells, impaired sperm production, and abnormalities in elongation, resulting in reduced fertility [42,43,44]. However, AGO-2-deficient mice do not demonstrate altered spermatogenesis [43]. The inactivation of DROSHA in spermatogenic cells within postnatal mice testes induces a severe depletion of germ cells, including low counts of elongating spermatids [45]. DICER is also essential for Sertoli cell survival. The loss of DICER in Sertoli cells is associated with high levels of apoptosis in those cells and impaired spermatogenesis, leading to azoospermia [46,47].

A number of miRNAs expressed in different cell types involved in spermatogenesis and their roles have been studied and are listed in Table 1. A highly regulated balance between self-renewal and differentiation of SSCs is essential to preserving spermatogenesis from puberty throughout life [6]. Several factors are involved in the maintenance of a sufficient stem cell population, protecting from SSC exhaustion due to unnecessary differentiation or excessive production of SSCs [48]. These factors are mostly produced by the Sertoli and germ cells [48]. However, other factors produced by Leydig cells, which form the interstitial tissue, and hormones, such as testosterone, can influence Sertoli cell function [48]. The role of miRNAs in regulating this precise balance has been described in various studies [6]. Among others, Glial cell line-derived neurotrophic factor (GDNF), fibroblast growth factor (FGF) secreted by Sertoli cells, or B cell CLL/lymphoma 6 member (BCLB6) protein have been identified as essential factors for maintaining the SSC pool or self-renewal [49]. GDNF functions by targeting multiple signaling pathways, particularly via a receptor formed by the Ret tyrosine kinase and a ligand-specific co-receptor GFR*α*1 [33]. BCLB6 is one of the targets of GDNF [49]. Moreover, the promyelocytic leukemia zinc finger (PLZF) protein, which is expressed in SSCs [48], directly represses the transcription of the receptor c-Kit, which is known to be important in the transition of spermatogonia [49]. By preventing the differentiation of spermatogonia, PLZF also allows for the maintenance of self-renewal and, thus, the pool of SSCs [49]. The Ets-related molecule, ETV5, expressed in both Sertoli and germ cells, plays a role in the maintenance of the SSC niche, which is the microenvironment where SSCs are located [33,49].

In contrast, KIT-ligand, which is produced by Sertoli cells, as well as notch signaling, both promote cell differentiation [17]. In addition, retinoic acid (RA), an active metabolite of vitamin A, is also involved in spermatogonial differentiation [6]. It has been suggested that RA plays its role by targeting several genes encoding proteins involved in spermatogenesis and promoting differentiation, such as stimulated by retinoic acid gene 8 *(Stra8)*, Kit proto-oncogene Receptor Tyrosine Kinase *(Kit)*, and Cyclin D2 *(Ccnd2)*, but the underlying mechanism needs to be further investigated [50,51].

**Table 1 ijms-25-00460-t001:** miRNAs in spermatogenesis, their functions, and target genes. *C-Kit*: Kit proto-oncogene receptor tyrosine kinase; *Stat3*: Signal inducer and activator of transcription 3; *Bim*: BCL-2 interacting mediator of cell death; *Col1a2*: Collagen type 1 alpha 2 chain; *Ccnd1*: Cyclin D1; *Plzf*: promyelocytic leukemia zinc finger; *FoxO1*: Forkhead Box O1; *Med1*: Mediator complex subunit 1; *Rbfox2*: FOX-1 homolog 2; *Stra8*: stimulated by retinoic acid gene 8 *(Stra8); Dmrt6*: Double sex-related gene *Dmrt6; Dmrt1*: double sex and Mab-3-related transcription factor 1; *Gfrα1*: GDNF family receptor 1; *Jazf1*: Zinc finger 1; *ATF1*: activating transcription factor 1; *Nanos2*: Nanos C2HC type zinc finger; *E2F-pRb*: E2 factor-retinoblastoma protein pathway; *Tnp2*: Transition protein 2; *Prm2*: protamine 2; *Hsf2*: Heat shock factor 2; *Gli3*: GLI family zinc finger 3; *Lrp6*: low density lipoprotein receptor-related protein 6; *FoxD1*: forkhead/winged-helix; *Dsc1*: desmocollin; *Pafah1b1*: platelet-activating factor acetylhydrolase 1b regulatory subunit 1 gene; *Star*: Steroidogenic acute regulatory protein; *Sf1*: steroidogenic factor-1 gene.

microRNA	Cell Type	Function and/or Targeted Genes in Spermatogenesis	Model	References
miR-17-92 cluster	Primordial germ cells Undifferentiated spermatogonia	Promotes survival and proliferation of PGCs Promotes stem cell self-renewal Targets *Stat3*, *Bim*	Mouse	[43] [52] [53]
miR-290-295 cluster	Primordial germ cells	Role in embryonic development Promotes germ cell migration	Mouse	[43] [54]
Let-7 family (7a,7b,7c,7d and 7e)	Spermatogonia Spermatocytes	Promotes spermatogonial differentiation Targets *Mycn*, *Col1a2*, *Ccnd1*	Mouse	[50]
miR-125a	Primordial germ cells	Targets *Lin28*	Mouse	[43]
miR-9	Primordial germ cells	Targets *Lin28*	Mouse	[43]
miR-26b	Undifferentiated spermatogonia	Promotes spermatogonial differentiation Targets *Plzf*	Mouse	[55]
miR-135	Undifferentiated spermatogonia	Controls stem cell self-renewal Targets *FoxO1*	Mouse	[56]
miR-146a	Undifferentiated spermatogonia	Promotes stem cell self-renewal, blocks the retinoic acid-induced differentiation of SSCs Targets *Med1*	Mouse	[57]
miR-21	Undifferentiated spermatogonia	Promotes stem cell self-renewal GDNF pathway	Mouse	[58]
miR-20 and miR-106a	Undifferentiated spermatogonia	Promotes stem cell self-renewal Targets *Stat 3*, *Ccnd1*	Mouse	[59]
miR-202	Undifferentiated spermatogonia	Promotes stem cell self-renewal Targets *Rbfox2*, *Stra8*, *Dmrt6*	Mouse	[60] [61]
miR-100	Undifferentiated spermatogonia	Promotes stem cell self-renewal Targets *Stat3* *(indirectly)*	Mouse	[62]
miR-221/222	Undifferentiated spermatogonia	Promotes stem cell self-renewal Targets *Kit*	Mouse	[63]
miR-224	Undifferentiated spermatogonia	Promotes stem cell self-renewal +: *Plzf*, *Gfrα1* −: *Dmrt1*	Mouse	[64]
miR-322	Undifferentiated spermatogonia	Promotes stem cell self-renewal Targets *Rassf8* +: *Gfrα1*, *Etv5*, *Plzf* −: *Stra8*, *C-Kit*, *Bcl6*	Mouse	[39]
miR-31-5p	Undifferentiated spermatogonia	Regulates proliferation and apoptosis of SSCs Targets *Jazf1*	Human	[65]
miR-122-5p	Undifferentiated spermatogonia	Stimulates proliferation, inhibits apoptosis	Human	[66]
miR-34c	Undifferentiated spermatogonia Spermatocytes Round spermatids	Promotes differentiation Targets *ATF1*, *Nanos2*	Mouse	[67] [68,69,70,71]
miR-449	Spermatocytes Spermatids	Promotes differentiation Targets E2F-pRb pathway	Mouse	[72]
miR-469	Spermatocytes Round spermatids	Chromatin compaction and condensation Targets mRNA *Tnp2* and *Prm2*	Mouse	[73]
miR-122a	Round spermatids	Chromatin compaction and condensation Targets *Tnp2*	Mouse	[74]
miR-18a	Spermatocytes	Chromatin compaction and condensation Targets *Hsf2*	Mouse	[75]
miR-133b	Sertoli cells	Promotes proliferation of SCs Targets *Gli3*	Human	[76]
miR-202-3p	Sertoli cells	Regulates proliferation of SCs Targets *Lrp6* and *Ccnd1*	Human	[77]
miR-471-5p miR-463 miR-201	Sertoli cells	Hormone-responsive Regulates the expression of *FoxD1* and *Dsc1* Testis specific	Mouse	[78] [79]
miR-181-c/d	Sertoli cells	Regulates proliferation of SCs Targets *Pafah1b1*	Mouse	[80]
miR-320-3p	Sertoli cells	Regulates germ cell support	Mouse	[81]
miR-382-3p	Sertoli cells	Its decline at puberty is correlated with the onset of spermatogenesis	Mouse	[82]
miR-140-3p/5p	Leydig cells	Differentiation of male gonads	Mouse	[83]
miR-150	Leydig cells	Regulation of steroidogenesis Targets *Star*	Mouse	[84]
miR-300-3p	Leydig cells	LH regulation Targets *Sf1*, *FoxO1*	Mouse	[85]

Among the numerous miRNAs studied, miR-146 has been shown to be highly expressed in undifferentiated mouse spermatogonia compared to differentiating spermatogonia. Its overexpression has been found to inhibit RA-mediated differentiation of spermatogonia. MiR-146 has been demonstrated to bind to the 3′-UTR region of the mediator complex subunit 1, *Med1*, a coregulator of RA-receptors [57]. This leads to a decrease in *Med1* transcript levels, as well as *Kit* oncogene transcript levels, to regulate the effects of RA on spermatogonial differentiation. MiR-221/222 expression levels are also downregulated in undifferentiated spermatogonia exposed to RA, while their overexpression makes undifferentiated spermatogonia resistant to RA-induced transition and incapable of differentiation in vivo [63]. The downregulation of these miRNAs leads to the expression of the KIT receptor and, thus, the transition from KIT^−^ to KIT^+^ spermatogonia. By repressing KIT expression, miR-221/222 appears to play a crucial role in maintaining the undifferentiated state of spermatogonia [63]. On the other hand, Tu et al. demonstrated an increase in miR-26b expression in undifferentiated spermatogonia treated with RA. Their results demonstrate that miR-26b aids the transition from KIT^−^ to KIT^+^ mouse spermatogonia by directly targeting PLZF, which is known to normally promote self-renewal [55]. MiR-202-3p, which is also highly expressed in mouse spermatogonial stem cells, has been demonstrated to be regulated either by GNDF, which increases its expression, or by RA, which acts as a down-regulator. Knock-out of MiR-202 in cultured SSCs leads to premature differentiation [60]. Furthermore, transient miR-21 inhibition in SSC-enriched germ cell cultures results in an increase in germ cell apoptosis, indicating that miR-21 has a role to play in the maintenance of the SSC population. Moreover, Niu et al. demonstrated that miR-21 is regulated by the transcription factor ETV5, which is known to be critical for SSC self-renewal [58]. Both miR-20 and miR-106a, which are preferentially expressed in SSCs, have also been identified as essential to the promotion of self-renewal of SSCs. The target genes of these miRNAs include *Stat3* and *Ccnd1*, and inhibition of miR-20 and miR-106a promotes self-renewal [59]. Finally, miR-322, which is also highly expressed in SSCs, not only targets *Rassf8* (ras association domain family 8) but increases *GFRa1*, *Etv5*, and *Plzf* mRNA expression while decreasing the expression of differentiation-promoting factors such as *Stra8*, *C-Kit*, and *Bcl6*. Together, this promotes self-renewal and inhibits the differentiation of SSCs [39]. MiR-18a, miR-122a, and miR-469 have been described in spermatocytes and round spermatids for their involvement in the process of chromatin compaction and condensation [73,74,75].

In addition to their expression in germ cells, several miRNAs have been identified in Sertoli and Leydig cells, where they regulate the activity of these somatic cells [76,77,78,79,80,81,82,83,84,85].

## 4. MiRNAs in Male Infertility and Their Use as Biomarkers

Several infectious and non-infectious pathophysiological processes can affect the male reproductive organs and lead to infertility. Infertility is defined as the inability to achieve a spontaneous pregnancy after one year of regular unprotected sexual intercourse [86]. Approximately half of infertility cases are attributed to male factors, while male infertility etiology remains unexplained in more than 50% of cases [87,88]. Spermatogenesis is a complex process with multiple steps, and dysregulation at any of these may cause male infertility. Fertility-related issues can lead to alterations in the expression profile of miRNAs, and this dysregulation may be a contributing factor to infertility [37]. Several studies have reported altered miRNA expression profiles with an infertility phenotype, and, therefore, miRNAs could have the potential to serve as diagnostic biomarkers (reviewed in [36,37,38,89]). Table 2 summarizes a number of studies that demonstrate the link between miRNA profile and impaired fertility. MiRNAs are also endogenous molecules that can be expressed in extracellular fluids and have been detected in seminal plasma, where their expression levels remain stable across multiple samples from the same individual [90]. Several studies have shown that miRNAs can be up- or down-regulated in human seminal plasma from patients with azoospermia, oligozoospermia, and asthenozoospermia [90,91,92,93,94,95]. Recently, miRNA expression patterns in urine and semen samples from non-obstructive azoospermia (NOA) patients were evaluated as potential biomarkers and proved to be effective for predicting the presence of testicular spermatozoa and spermatogonia before testicular sperm extraction (TESE) [96]. The analysis of miRNA expression profiles in testicular tissue identified different miRNAs that are differentially expressed in NOA or obstructive azoospermia (OA) patients and could potentially be used as diagnostic biomarkers. Examples include miR-10b-3p and miR-34b-5p, which could serve as predictive biomarkers of azoospermia, as well as miR-122-5p, which was found to be up-regulated in human spermatogonia from OA patients compared to NOA patients [66,97].

Improving the diagnosis of idiopathic male infertility with the help of molecular biomarkers combined with semen analysis and/or histopathologic diagnosis would be a major advancement in the field of male infertility management. MiRNAs appear to meet numerous requirements to be valuable biomarkers: they are present in various bodily fluids, they have been shown to be stable in seminal plasma [90], and appear prior to proteins and, thus, could aid the early detection of disease. Despite encouraging results, it is important to mention that there are still limitations that need to be addressed before miRNAs can be implemented into clinical practice. First, the sample sizes were often relatively small in the above-mentioned studies. Moreover, it is important to mention the lack of well-established and standardized procedures regarding sample collection, storage, analysis, and normalization of miRNA expression data [90,101]. Indeed, the identification of miRNAs and their target genes is based on the use of bioinformatics techniques that have their own limitations, as each algorithm is associated with false-positive and false-negative prediction rates, as well as providing speculative results that need further experimentation to be validated [90]. The validation techniques also need to be improved and to be tested on larger populations. In addition, despite the fact that miRNAs offer the opportunity for multiple targeting, their targets and functions should be thoroughly investigated with regard to the different tissues or organs where miRNAs are expressed [101]. Moreover, the genetic variability of each individual can be another limitation to the clinical application of these biomarkers.

Additionally, dysregulation of miRNAs has been associated with the development of several cancers, including testicular cancers [102]. MiRNAs can function as either oncogenes or tumor suppressors. For example, one study described the role of miR-371-373 cluster as an oncogene in the development of testicular germ-cell cancer via its interference with the p53 pathway [99]. Gillis et al. showed that miRNA expression profiles could also help to distinguish between type II and type III germ cell tumors and highlighted several miRNAs as tumor-specific [100]. Furthermore, the involvement of miRNAs in prostate cancer has been extensively explored in multiple studies [103]. The differences in miRNA expression profiles between healthy and tumoral tissues hold the potential for enhancing diagnosis and prognosis in the field of oncology. Nevertheless, there are still some clinical limitations in this area that should be mentioned. In particular, testicular cancer (TC) management remains challenging, both in terms of early diagnosis and monitoring, and numerous studies (reviewed in [104]) have assessed the ability of miRNAs to work as biomarkers in TC at various levels. However, controversial results on the ability of miRNAs to work as early diagnostic markers of the disease have been reported. Therefore, to translate from the numerous preclinical studies into the clinical development of cost-effective miRNA-based approaches as a tool in the diagnosis and follow-up of TC, more clinical trials should be conducted, and further research is needed before miRNAs can be implemented into clinical practice [104,105].

In addition, infectious conditions can result in erectile dysfunction, infertility, or cancers of the male reproductive tract [37,99,106,107]. To date, a limited amount of data exists concerning modifications in miRNA expression profiles following infection or inflammation of the male reproductive organs or reproductive tract. In this regard, Parker et al. studied miRNA expression profiles following immune activation due to inflammation in rat testes in order to identify miRNAs that target inflammatory genes [106]. Indeed, the regulation of inflammatory gene expression by miRNAs has been suggested as a potential control mechanism by the male reproductive system to maintain immune stability [106]. The results of this study showed downregulation of miRNAs that target inflammatory genes. Some of these miRNAs, such as miR-34c or miR-449, were already recognized as essential for germ cell survival and differentiation during physiological spermatogenesis [67,72]. Therefore, the authors suggested that immune system activation of the testis following infectious insult might lead to several pathologies by downregulation of miRNAs essential to normal testis function [106]. However, to induce an inflammatory reaction, the authors opted for bacterial antigen lipopolysaccharide (LPS) treatment, which is known to reduce androgen production. It cannot be ruled out that the downregulation of some miRNAs could be androgen-dependent rather than having their expression modulated by the inflammatory response [106]. Furthermore, the androgen receptor can also be a target of miRNAs. Cytokines have been shown to participate, with androgens, in the maintenance of testicular function, including spermatogenesis. Additionally, cytokines play a crucial role in the testicular immune system [107].

Numerous aspects related to the unique immunogenic environment of the testes and immune response in pathophysiological conditions remain to be elucidated, including the role miRNAs play in this context.

## 5. Principles of miRNA-Based Therapeutics

Exploiting the natural regulatory function of miRNAs, miRNA-based therapeutics aim to modulate gene expression patterns and restore aberrant gene expression. Generally, miRNA-based therapeutic strategies are divided into miRNA replacement and inhibition approaches [105]. MiRNA replacement strategies use miRNA mimic molecules, which are double-stranded small RNAs that aim to restore or up-regulate the expression of endogenous miRNA targets. Strategies that focus on inhibition of miRNA function are based on the use of miRNA antagonists or anti-miRNAs, which are antisense single-stranded oligonucleotides (ASOs) with a sequence complementary to the mature form of the miRNA target [108]. To date, several clinical studies registered at clinicaltrials.gov for miRNA biomarkers have been completed, while clinical trials for miRNA therapeutics are active, but none have reached phase III yet [109]. The main challenges for the transition of miRNA therapeutics from the bench to the bedside are the development of miRNA delivery systems and immune-related side effects.

## 6. MiRNAs in Fertility Preservation Strategies

### 6.1. Semen Cryopreservation

Cryopreservation is a commonly used technique for long-term storage of sperm samples, but it can induce damage to spermatozoa, affecting their quality and leading to impaired fertilization capacity [110]. The precise mechanisms underlying cryo-induced DNA damage are complex and multifactorial, involving factors such as ice crystal formation, osmotic stress, oxidative stress, and altered DNA packaging that can induce genomic and epigenomic instability [111,112]. The most frequent DNA damage that occurs in the male gamete during cryopreservation is DNA fragmentation [113]. According to several studies, increased caspase activity triggering apoptosis can result from cryo-injury [114,115]. However, the use of caspase inhibitors in a cryoprotectant medium does not reverse this effect [116]. Emerging evidence also suggests that cryopreservation results in changes in miRNA expression, affecting sperm quality. A study in boar sperm revealed that 23 miRNAs were differentially expressed during cryopreservation, and their function was associated with energy metabolism, sperm structure, motility, and apoptosis [117]. Moreover, it has been demonstrated that cryopreservation is associated with decreasing expression of P53, miR-43c, and miR-184 in human sperm, leading to altered sperm motility, DNA fragmentation, and fertilization capacity [115]. Another study in mice and human spermatozoa revealed that a number of miRNAs implicated in the fertilization process are downregulated after cryopreservation, particularly miR-148b-3p, by targeting Pten, which plays a key-role in the PI3K/AKT signaling pathway [118]. Paternal miRNAs seem to play a crucial role in the early embryonic developmental process by targeting maternal mRNAs and affecting the activation of zygotic genes [119]. It is also interesting to mention that a recently discovered miRNA family, called CryomiRs, acts as modulators of freeze tolerance and metabolism in mammalian hibernators [120]. Given that some of the advances in cryopreservation techniques have been derived from mimicking the metabolic adaptations used by freeze-tolerant animals, CryomiRs could open the way for discovery of new miRNAs with cryoprotective properties [121]. MiRNAs can exert protective effects against cryo-induced cellular damage due to their ability to fine-tune multiple pathways, including DNA damage response, apoptosis, and oxidative stress, by targeting specific genes involved in these pathways. Understanding the role of miRNAs in cryo-induced DNA damage can provide valuable insights into the underlying molecular mechanisms and potentially guide the development of strategies to mitigate the harmful effects of cryopreservation on sperm DNA [122].

### 6.2. Stem Cell-Based Fertility Preservation

The application of stem cell therapy using SSCs has been proposed as a promising approach to fertility preservation and restoration strategies, especially for prepubertal cancer patients receiving gonadotoxic therapy [123]. Back in 1994, autotransplantation of SSCs into the testes was used to successfully restore spermatogenesis in mice as well as in non-human primates rendered infertile with alkylating chemotherapy [124,125]. The clinical application of this approach requires the injection of SSCs, isolated from cryopreserved immature testicular tissue, into the seminiferous tubules of the testes. Despite the encouraging results in animal models, the only reported clinical study in humans was not successful, and questions remain about the safety of this approach as there is a risk of residual disease. Moreover, in pediatric cancer patients, the time period between tissue harvesting and fertility treatment is long, and thus, genetic and epigenetic abnormalities may occur that affect SSC quality [126]. During the cryopreservation period, SSCs are susceptible to damage caused by reactive oxygen species (ROS); therefore, agents with antioxidant properties have been used to prevent this damage. Recently, it has been proposed that miRNAs with anti-apoptotic and antioxidant activities can protect SSCs from cryopreservation-induced injury. For example, it has been shown that the in vitro transfer of miR-30a-5p mimic–liposomic conjugations in mouse SSCs before the freeze–thaw process exhibits anti-apoptotic and antioxidant properties that lead to increased viability, number, and diameter of the SSC colonies [127]. Therefore, the introduction of miRNAs into cryopreservation protocols may maintain or even improve SSC quality before transplantation. Moreover, it has been proposed that SSC transplantation could be improved by mesenchymal stem cell (MSC) co-transplantation [128]. MSCs produce paracrine factors that may help to restore the niche cells (Sertoli, Leydig, peritubular cells) that are also affected by oncological treatments [129]. The regenerative properties of MSCs improve the efficacy of SSC transplantation, but there is a concern regarding epigenetic modifications in the offspring. Characterization and analysis of the MSC secretome might help to elucidate the precise role of MSCs in tissue regeneration. For example, it has been demonstrated that MSCs release exosomes containing different RNA molecules, such as miRNAs and proteins, that may be key mediators of this process [130,131]. According to a recent study, MSC treatment mitigated busulfan-induced azoospermia in rats by altering the expression of spermatogenesis-related miRNAs and their target genes [132]. The elevated steroid hormone levels and increased expression of germ cell-specific genes indicated the beneficial effect of MSC-derived miRNAs. Therefore, it would be interesting to identify MSC-derived miRNAs and explore their role in manipulating the microenvironment surrounding transplanted SSCs, promoting their interaction with niche cells and optimizing the regeneration of spermatogenesis.

### 6.3. MiRNA-Based Pharmacoprotective Approaches

The use of miRNAs as pharmacological agents holds great promise in preserving male fertility during oncological treatment, potentially offering a novel therapeutic strategy to safeguard reproductive capacity and improve the quality of life for cancer survivors.

The pharmacological protection of the testis from gonadotoxic agents is particularly attractive for patients and prepubertal boys who cannot benefit from semen cryopreservation. GnRH analogs, granulocyte colony-stimulating factor (G-CSF), and melatonin have been investigated as protective-restorative therapies with variable results. The idea for the use of miRNAs in male fertility preservation strategies stems from their key role as regulators of essential cellular gene expression pathways affected by oncological treatments. This alternative therapeutic approach has been tested in female fertility preservation strategies, where it has been proven to be effective for regulating apoptosis in mouse ovaries exposed to alkylating agents in vitro. In this experimental model, the downregulated expression of let-7a following exposure to the active metabolite of cyclophosphamide, 4HC, was restored using let-7a mimic transfection approaches, demonstrating fertility preservation properties. Similarly, the application of miRNA restoration or reduction approaches to prepubertal testes could represent an innovative strategy to reduce cell arrest, DNA damage, and apoptosis induced by chemotherapy in immature germ cells.

### 6.4. Limitations

While the exploration of miRNAs in semen cryopreservation and stem cell-based fertility preservation shows promise, it is essential to critically assess the existing limitations and safety concerns associated with these approaches. Several potential hurdles and controversies exist. First, the application of miRNAs in fertility preservation is still largely experimental, with no clinical evidence supporting their efficacy and safety in human subjects. The identification of miRNAs with fertility preservation potential and the launch of subsequent clinical trials are needed to establish their reliability. Additionally, miRNAs, known for having multiple targets, may result in off-target toxicity by influencing genes and pathways beyond the initial goal. Understanding and mitigating these possible side effects is essential to assuring the precision and safety of miRNA-based interventions. In addition, potential impacts on preserved gametes or stem cells exposed to exogenous miRNAs need to be thoroughly investigated in order to avoid any possibility of teratogenesis.

## 7. miRNA Therapeutics: Limitations and Solutions

The use of naked miRNAs in restoration or inhibition approaches poses a number of limitations [133]. MiRNAs are negatively charged nucleic acids that interact with positively charged cellular membranes, leading to limited cellular uptake. Moreover, naked miRNAs have a short half-life as they are unstable molecules and are prone to degradation by nucleases found in the systemic circulation. This rapid degradation can limit their bioavailability and therapeutic effects. Additionally, miRNAs may exhibit off-target effects, as they can potentially bind to unintended mRNA targets, leading to undesirable alterations in gene expression. Also, immune responses triggered by the presence of exogenous miRNAs can limit their efficacy and raise safety concerns.

Modifications of the miRNA backbone, such as conjugation with cholesterol moieties, incorporation of chemically modified nucleotides such as 2′-O-methyl into locked nucleic acids (LNAs), miRNA sponges, and others that have been described in recent reviews, can increase the stability of synthetic miRNAs and enhance cellular uptake [134,135,136]. Recently, the use of encapsulation strategies has been investigated to protect miRNAs from degradation and facilitate their targeted delivery to specific cells or tissues. These modifications of synthetic miRNAs hold great potential for enhancing the stability, cellular uptake, and therapeutic efficacy of these molecules, paving the way for their successful translation into clinical applications.

Currently, the integration of nanotechnology into male fertility preservation has the potential to revolutionize the field, providing more effective and personalized approaches to preserve male reproductive function and improve outcomes for individuals facing fertility challenges. Several types of nanoparticles have been investigated for the delivery of miRNAs in the landscape of cancer treatment, and their potential application in male fertility preservation is also very attractive [137] (Figure 3). Lipid-based nanoparticles, such as liposomes and lipid nanoparticles (LNPs), have shown promise in mRNA/miRNA delivery due to their biocompatibility and ability to encapsulate and protect RNAs from degradation [138,139]. It has been shown that these lipid-based nanoparticles can be further modified with cholesterol, amino groups, and phosphate linkers (CAPs) to enhance delivery into spermatocytes after testicular injection [140]. In addition, polymeric nanoparticles, such as poly (lactic-co-glycolic acid) (PLGA) nanoparticles, offer advantages such as controlled release, stability, and the ability to encapsulate miRNAs [141]. Polyethyleneimine (PEI) nanoparticles, which have a high positive charge and can condense miRNAs, have also been explored for efficient miRNA delivery [142]. Moreover, inorganic nanoparticles, including gold nanoparticles (AuNPs) and silica nanoparticles, have gained attention for their unique physicochemical properties and surface modifications for targeted delivery. The small size, stability, and ease of surface functionalization of AuNPs make them particularly interesting for biomedical applications. In the context of miRNA delivery, AuNPs can be functionalized with specific ligands or coatings to enable targeted and efficient delivery of miRNAs to desired cells or tissues [143,144,145]. In female fertility preservation, AuNPs have been proposed as vehicles for the delivery of specific miRNAs to the ovary to modulate gene expression and restore normal cellular functions [146]. Likewise, the surface functionalization of AuNPs with cell-penetrating peptides or other targeting ligands can enhance their cellular uptake and promote selective delivery to desired cell populations, such as spermatogonial stem cells or germ cells. By utilizing AuNPs as miRNA nanocarriers, it is possible to achieve controlled and sustained release of miRNAs, ensuring their optimal therapeutic effects.

The discovery of extracellular vesicles (EVs), such as micro-vesicles, exosomes, and other subcategories, has brought to light an alternative pathway of intercellular communication, regulating a variety of physiological and pathological processes. EVs are lipid bilayer particles secreted by cells that mediate the horizontal transfer of key molecules such as DNA, RNA, small non-coding RNAs (sncRNA, miRNAs), proteins, and lipids to recipient cells [147]. In the context of human reproduction, EVs play a crucial role in facilitating cellular interactions and modulating reproductive processes. For example, it has been demonstrated that epididymal epithelial cells contribute to the maturation process of spermatozoa through the secretion of EVs known as epididymosomes [148]. Studies in animal models have shown that epididymosomes contain proteins and a plethora of miRNAs that can modify the sperm proteome and epigenome, respectively, and regulate post-testicular sperm maturation [149,150]. Additionally, EVs released by prostate epithelial cells into prostatic fluid, known as prostasomes, appear to play an important role in the acquisition of sperm fertilizing capacity. Specifically, prostasomes modulate sperm motility, capacitation, and induction of the acrosome reaction while they protect the sperm from the female immune system, facilitating endometrial implantation [151,152]. Given that different kinds of EVs promote reproductive success, they are interesting candidates for miRNA nanocarriers in male fertility preservation strategies to protect sperm during gonadotoxic treatments. Isolated seminal EVs (epididymosomes, prostasomes) or tailored engineered nanovesicles resembling naturally derived EVs can be loaded with the selected miRNA(s) by co-incubation or electroporation [153]. Naturally derived EVs may offer the opportunity for selective targeting through surface interactions facilitated by adhesion proteins, tetraspanins, or integrins, leading to recipient cell recognition and miRNA internalization [154]. The molecular characterization of EV surface markers will be useful for the creation of artificial EVs for targeted miRNA delivery. Despite the fact that cell line-derived EVs are not immunogenic and toxic, the in vivo administration and clinical application of these is still challenging and needs further investigation [155]. Clinical trials related to exosome-based therapeutic delivery systems are ongoing, but several limitations and questions have arisen [156]. One of the main challenges is the possibility of endosomal trapping that leads EVs to lysosomes and cargo degradation. Nevertheless, there are lysosomotropic endosomal escape-enhancing compounds that can be used to modify EVs and bypass the lysosomal pathway [157]. In addition, the conventional methods of EV isolation are complex and high-cost processes with suboptimal results, but recently, new methodologies such as microfluidics have opened new horizons in recovering, analyzing, and quantifying EVs, facilitating their use in novel approaches to male infertility [158].

## 8. MiRNA Delivery to the Testis

Successful in vivo nucleic acid delivery to the testes has been described using the injection of a plasmid followed by electroporation. The procedure resulted in the restoration of complete spermatogenesis in ring finger protein 20 knock-out (Rnf20 KO) mice [159]. Similarly, nucleic acids have been transferred into germinal cells, Sertoli cells, and Leydig cells by transfection with different adeno-associated virus (AAV) serotypes [160,161]. Recently, transitory rescue of defective spermatogenesis was achieved by the transfection of naked mRNA and the temporary expression of the missing protein in a genetic defect associated with infertility [162]. Capped and poly-tailed mRNAs were injected in the rete testis of mice, and transfection was induced by electroporation. Du et al. restored the expression of DNA meiotic recombinase 1 (DMC1) in Dmc1 KO mice by the injection of a self-amplifying RNA-Dmc1 into the testis [140]. These studies demonstrate that the delivery of nucleic acid to germinal cells is feasible and could be extended to small RNAs, such as miRNAs, keeping in mind the need to develop biotechnological tools to improve RNA stability, cellular targeting, and transfection.

Determining the optimal route of clinical administration for nanoparticles into the testes is crucial to ensuring effective and safe delivery of miRNAs in male fertility preservation (Figure 3). Currently, several routes have been explored in animal models, each with its advantages and limitations. One approach is the intratesticular (IT) injection of nanoparticles into the interstitial compartment of the testes and rete testes (RT) injection into the adluminal compartment of the seminiferous tubules. This allows precise and localized delivery, enabling efficient uptake of nanoparticles by the testicular cells while minimizing systemic exposure. However, this route requires invasive procedures and may pose certain risks, such as infection or tissue damage. According to a recent study, the authors used IT-AuNP injection, but AuNPs could not penetrate through the blood–testis barrier (BTB), and no reproductive toxicity was observed in mice. However, when they used RT-AuNP injection, AuNPs were transferred from the adluminal to the interstitial compartment via Sertoli cells, resulting in testicular damage and inflammatory response [163].

Another possible route is the systemic administration of nanoparticles, such as intravenous injection, which allows for wider distribution and systemic circulation. Systemic administration can facilitate the delivery of nanoparticles to the testes via passive targeting or surface modifications that enhance their accumulation in the testicular tissue. However, systemic administration may result in lower nanoparticle accumulation in the testes due to clearance mechanisms and limited penetration through the BTB. The BTB protects the spermatogenesis process by providing an immunoprotective function while it creates a balanced environment of ions and molecules in the seminiferous tubules, but its presence makes the delivery of nanoparticles to the germ cells and gametes more difficult [164]. Additionally, nanoparticles may accumulate in other organs, potentially leading to off-target effects. Further research is needed to optimize the route of administration, considering factors such as nanoparticle properties, desired therapeutic outcomes, and patient-specific considerations. The ideal route should ensure efficient nanoparticle delivery to the testes while minimizing invasiveness, toxicity, and systemic effects, ultimately maximizing the therapeutic potential of miRNA-loaded nanoparticles in male fertility preservation.

## 9. Future Perspectives

The use of miRNAs as a fertility preservation strategy in males is a promising area of research; however, it is important to acknowledge a number of limitations associated with this approach. First, miRNAs are small, non-coding RNA molecules that regulate gene expression and play a crucial role in various biological processes, including spermatogenesis. Our understanding of their precise functions and interactions within the male reproductive system is still evolving. The complexity of miRNA networks and their dynamic nature pose challenges in identifying specific miRNAs that can effectively promote spermatogenesis or protect against fertility decline. Additionally, the delivery of exogenous miRNAs to the testes remains a significant hurdle. Efficient and targeted delivery systems that can ensure the uptake and sustained expression of miRNAs in the male germline need to be developed. Furthermore, the long-term safety and potential off-target effects of manipulating miRNA expression in the male reproductive system need to be thoroughly investigated. Given these limitations, further research is required to fully elucidate the potential of miRNAs as a reliable and practical fertility preservation strategy in males.

## Figures and Tables

**Figure 1 ijms-25-00460-f001:**
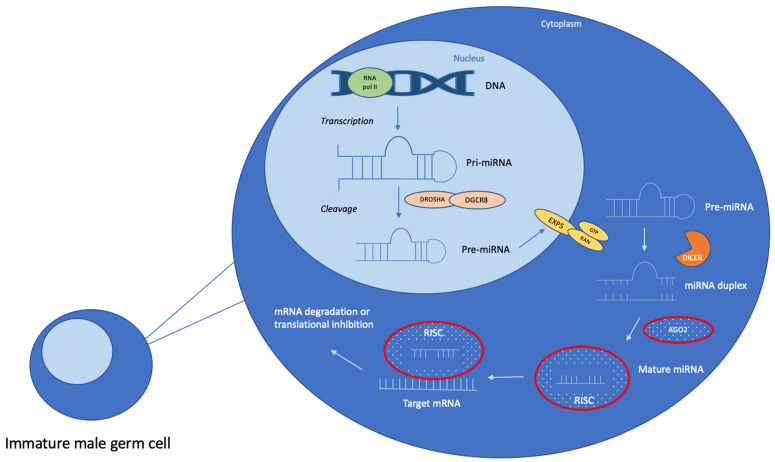
The canonical miRNA biogenesis pathway in immature male germ cells. In the nucleus, the miRNA coding region is first transcribed by an RNA polymerase II, which produces a pri-miRNA. It is then recognized by the DGCR8 protein, which is associated with an enzyme called DROSHA, forming a microprocessor complex. This complex cleaves the pri-miRNA into a pre-miRNA, which is exported outside the nucleus by Exportin-5. In the cytoplasm, another enzyme called DICER cleaves the pre-miRNA, leading to a miRNA duplex. It is charged into an argonaute protein (AGO); the passenger strand is degraded; and the guide strand, in combination with AGO, forms the RNA-induced silencing complex (RISC). The miRNA is now able to target genes.

**Figure 2 ijms-25-00460-f002:**
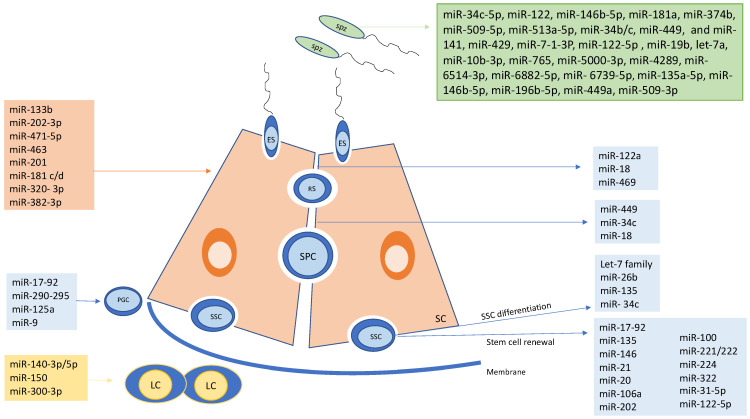
Schematic representation of miRNAs involved in the regulation of spermatogenesis. LC: Leydig cell; SC: Sertoli cell; SSC: spermatogonial stem cell; SPC: spermatocytes; RS: round spermatid; ES: elongated spermatid; PGC: primordial germ cell, Spz: spermatozoa.

**Figure 3 ijms-25-00460-f003:**
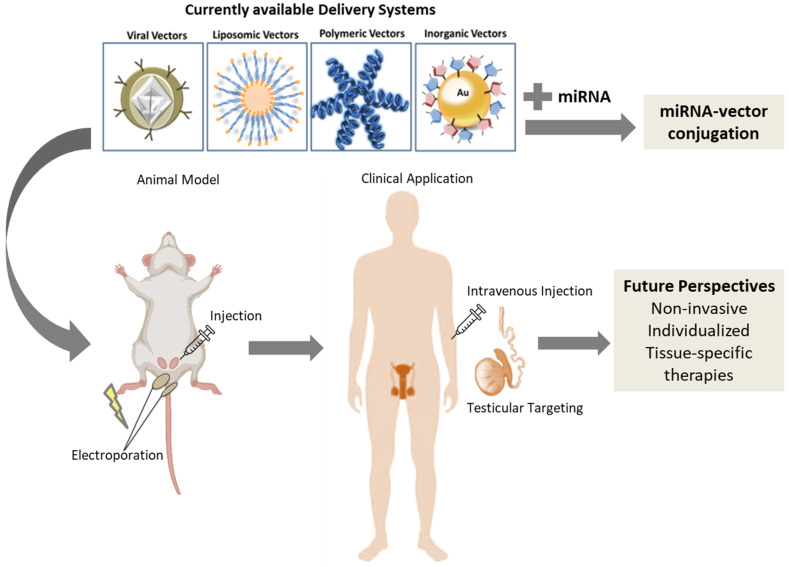
Proposed model for the in vivo administration of the miRNA-vector conjugation. At the preclinical level, the miRNA-vector conjugation (defined as a miRNA coupled with the delivery system) can be transferred by testicular/rete testis injection (and electroporation) into the mouse model. The testicular/rete testis administration route can be translated to human patients, but the ultimate preferred route would be the intravenous injection of the miRNA-vector conjugation combined with specific testicular targeting.

**Table 2 ijms-25-00460-t002:** Alteration in miRNA expression profile in male infertility and male genital cancers. NOA: non-obstructive azoospermia; qRT-PCR: quantitative real-time PCR; RNAseq: RNA sequencing.

Samples Used	Techniques	Down- or Up-Regulated miRNAs	Reference
Human seminal plasma	RNAseq and qRT-PCR validation	RT-qPCR: miR-34c-5p, miR-122, miR-146b-5p, miR-181a, miR-374b, miR-509-5p, and miR-513a-5p were down-regulated in patients with NOA and up-regulated in patients with asthenozoospermia	[91]
Human seminal plasma	qRT-PCR	miR-19b, let-7a were up-regulated in patients with NOA	[90]
Human seminal plasma	qRT-PCR (TaqMan)	miR-141, miR-429, and miR-7-1-3P were up-regulated in patients with NOA compared to control	[94]
Human semen sample Human purified spermatozoa and testicular biopsies	Microarray with RT-qPCR validation qRT-PCR	50 miRNAs up-regulated and 27 miRNAs down-regulated in asthenozoospermic patients compared to control. 42 miRNAs were up-regulated and 44 miRNAs down-regulated in oligoasthenozoospermic patients when compared with normozoospermic males. miR-34b*, miR-34b, miR-34c-5p, miR-122 down-regulated and miR-429 up-regulated in NOA and subfertile patients	[92] [98]
Human semen sample	qPCR	miR-34b-5p is down-regulated in patients with asthenozoospermia and oligozoospermia	[93]
Testicular tissue	Microarray and validation with qRT-PCR	129 miRNAs were differentially expressed in the NOA group compared to control Combination of miR-10b-3p and miR-34b-5p as a predictive biomarker of azoospermia	[97]
Human male germ cells isolated from testicular tissue	miRNA microarray	miR-122-5p is up-regulated in human spermatogonia of patients with OA compared to NOA patients	[66]
Human seminal plasma	High-throughput sequence technology and validation with RT-qPCR	6 miRNAs were up-regulated (miR-765, miR-5000-3p, miR-4289, miR-6514-3p, miR-6882-5p, miR-6739-5p) and 7 (miR-34b/c-5p, mR-135a-5p, miR-146b-5p, miR-196b-5p, miR-449a, miR-509-3p) were down-regulated in patients with asthenozoospermia compared to healthy men	[95]
Human seminal plasma and urine samples	Next-generation sequencing	13 miRNAs differentially expressed (10 in seminal plasma samples and 3 in urine) to predict the presence of testicular spermatogonia	[96]
Cell culture	Microarray Genetic screen	miR-372 and miR-373: potential novel oncogenes in testicular germ cell tumors by interfering with p53 pathway	[99]
Testicular tissue	qPCR	miRNA expression profile to distinguish type II and III germ cell tumors	[100]

## Data Availability

Not applicable.

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
