# Peer review of "The Role of microRNA in Spermatogenesis: Is There a Place for Fertility Preservation Innovation?"

_ijms, 2023, doi:10.3390/ijms25010460_

Round 1

Reviewer 1 Report

Comments and Suggestions for Authors

This study aims to envisages miRNAs as innovative treatment option in male fertility preservation-restoration strategies and describes the diverse applications where these new therapeutic tools can serve as fertiprotective agents.

I believe that the study has sufficient merit to be considered for publication on International Journal of Molecular Sciences, although major revisions are required.

MAJOR COMMENTS

-       Introduction: While the introduction outlines the importance of male fertility preservation, it lacks a clear statement of objectives and a roadmap for the article. Defining the scope and purpose early on would provide readers with a better understanding of what to expect. Additionally, incorporating specific examples or statistics on the prevalence of fertility issues in cancer patients would strengthen the introduction.  

-       3. The roles of miRNAs in spermatogenesis. While the section provides a comprehensive overview of miRNAs, it could benefit from a more explicit connection to the challenges outlined in the previous section. For instance, discussing how miRNAs specifically address the gonadotoxic effects of treatments would strengthen the narrative. Moreover, a critical analysis of the current understanding of miRNAs in spermatogenesis and their limitations in the context of cancer treatments is warranted. In this regard, I recommend this reference that I think are important and that can be of great help when modifying the manuscript (https://doi.org/10.3390/medicina59112033).

-       4. MiRNAs in male infertility and their use as biomarkers. This section establishes a link between miRNAs and male infertility, emphasizing their potential as diagnostic biomarkers. A critical discussion on the reliability and reproducibility of miRNA biomarkers in different patient cohorts or under various conditions is essential.

-       6. MiRNAs in fertility preservation strategies. While informative, this section would benefit from a more critical analysis of the existing limitations or safety concerns associated with using miRNAs in semen cryopreservation and stem cell-based fertility preservation. Addressing potential hurdles and controversies surrounding these approaches would provide a more nuanced perspective.

Comments on the Quality of English Language

Moderate editing of English language required

Reviewer 2 Report

Comments and Suggestions for Authors

The testicles and the male reproductive system create an environment that safeguards spermatogenic cells and sperm from immune attacks, despite these cells being highly immunogenic. Simultaneously, substances produced by the testicles, including androgens, impact the development and functions of the immune system. Immune system activation negatively affects both androgen and sperm production, making male fertility susceptible to compromise during systemic or local infections and inflammation.

Comments on the Quality of English Language

The review by Klees et al.,  contemplates the potential use of miRNAs as a novel treatment approach for preserving and restoring male fertility. It outlines various applications where these emerging therapeutic tools can act as agents to safeguard fertility. Overall, it is well-written and provides a comprehensive, up-to-date overview of the field.

Comments::

The immunological aspects of the putative roles of miRNAs need to be developed. Indeed, the testis and the male reproductive system create an environment that safeguards spermatogenic cells and sperm from immune attacks, despite these cells being highly immunogenic. Simultaneously, substances produced by the testicles, including androgens, impact the development and functions of the immune system. Immune system activation negatively affects both androgen and sperm production, making male fertility susceptible to compromise during systemic or local infections and inflammation. While reproductive biologists and immunologists have started to investigate the mechanisms governing these interactions, numerous essential details remain undiscovered. Therefore an update of the correlation of miRNNAs

Reviewer 3 Report

Comments and Suggestions for Authors

Klees et al. described the importance of miRNAs in spermatogenesis and the potential of using miRNAs for preserving male fertility, especially for those who undergo cancer treatment. They started by introducing how miRNAs are produced, their mechanism of function, and their roles in spermatogenesis. The authors then described how miRNAs could be used as biomarkers, and their utility in fertility preservation strategies, including semen cryopreservation, spermatogonia stem cell preservation, and miRNAs as pharmacological agents. Next, challenges and possible solutions to using miRNAs as therapeutics were discussed extensively, followed by strategies to efficiently deliver miRNAs into the testis.

This review article provides a well-organized description of how miRNAs can be utilized to overcome male fertility problems and improve reproductive health in patients, but also clearly points out areas with new knowledge and alternative approaches needed. This review article presents well summarized information regarding miRNAs in spermatogenesis and will be an important piece of work for the audience at the International Journal of Molecular Sciences. I have some comments below to hopefully strengthen this article:

1.     Please try to summarize or at least describe in a few sentences (preferably longer) near line 151 about the miRNAs that function in spermatogenesis, which you listed in Figure 2. Without mentioning about these miRNAs, the focus of this section (3: the roles of miRNAs in spermatogenesis) seems skewed. When readying this section without looking at the figure, I was wondering, why did the authors focused on molecules that are not miRNAs (which you spent quite a bit space on from line163 – line 190). An alternative is to describe and summarize on what you listed on table 1.

2.     Line 211, could you list what some of these potential diagnostic biomarkers are?

3.     Line 207, spell out “Non-obstructive azoospermia” here than later on line 210.

4.     Line 98: remove “be”

5.     Line 99: change “into” to “in”

6.     Line 100: change “into” to “by”

7.     Line 330 and 449: “hardness” is a noun. I assume you want to say it makes it harder for miRNAs to interact with positively charged membranes? Consider “hamper, hinder, impede”

8.     Line 333, change “prompted” to “prone”

9.     Line 338, miRNA backbone

10.  Line 340, remove “a”

11.  Figure 3 legend: please correct me but what is a miRNA-vector conjugation? Was this mentioned somewhere in the text or I just missed its meaning?

Comments on the Quality of English Language

minor grammar error.

Round 2

Reviewer 1 Report

Comments and Suggestions for Authors

I believe that the study has sufficient merit to be considered for publication

Reviewer 3 Report

Comments and Suggestions for Authors

The authors have addressed my comments properly.